# Temperature Relaxation in Glass-Forming Materials under Local Fast Laser Excitations during Laser-Induced Microstructuring

Alexander Minakov [1] and Christoph Schick [2,*]

1 Prokhorov General Physics Institute of the Russian Academy of Sciences, Vavilov Str. 38, 119991 Moscow, Russia; minakov@nsc.gpi.ru
2 Institute of Physics and Competence Centre CALOR, University of Rostock, Albert-Einstein-Str. 23–24, 18051 Rostock, Germany
* Correspondence: christoph.schick@uni-rostock.de; Tel.: +49-381-498-6880

**Featured Application: The obtained knowledge can be useful for understanding and optimizing various technologies with glass-forming materials under local fast laser excitations.**

**Abstract:** The ability to control the temperature distribution $T(t, r)$ and the rate of temperature change $\mathcal{R}(t, r)$ inside glasses is important for their microstructuring. The lattice temperature is considered at time $t$, exceeding the electron–phonon thermalization time, and at a distance $r$ from the center of the model spherical heating zone. In order to describe thermal excitations, the heat capacity of glasses must be considered as a function of time due to its long-term relaxation. A method for the analytical calculation of $T(t, r)$ and $\mathcal{R}(t, r)$ for glasses with dynamic heat capacity $c_{dyn}(t)$ is proposed. It is shown that during laser microstructuring, the local cooling rate $-\mathcal{R}(t, r)$ significantly depends on the time dispersion of $c_{dyn}(t)$. It has been established that at the periphery of the model heating zone of the laser beam focus, the local cooling rate can reach more than $10^{11}$ K/s. Strong cooling rate gradients were found at the periphery of the heating zone, affecting the microstructure of the material. This effect is significantly enhanced by the time dispersion of $c_{dyn}(t)$. The effect associated with this time dispersion is significant, even well above the glass transition temperature $T_g$, since even short relaxation times of the dynamic heat capacity $c_{dyn}(t)$ are significant.

**Keywords:** glasses; dynamic heat capacity; non-equilibrium heat transfer; laser-induced microstructuring; femtosecond laser processing; optical storage





## 1. Introduction

Glass-forming materials are often used to create optical elements and components for optical integrated circuits. In the last two decades, the technology for the femtosecond laser microfabrication of various optical structures inside glass-forming materials has been intensively developed. Due to the extremely high peak intensities at the focus of femtosecond laser pulses, nonlinear processes such as multiphoton absorption can lead to significant energy absorption even within transparent materials such as glass. Thus, the femtosecond laser-induced microstructuring of glass-forming materials opens up wide possibilities for various applications. Laser heating can cause irreversible local phenomena at the beam focus, such as phase transitions [1], the formation of bubbles or microfluidic channels [2–5], changes in chemical composition or refractive index [6,7] such as the diffusion and aggregation of silver ions [6], or photo-oxidation [7]. Spatially selective laser-induced crystallization or structure changes in glasses allows the direct recording of channel waveguides [1,8–14]. In fact, laser-written waveguides (with improved mode structures of guided light) can be fabricated through local laser heating inside a glass matrix [8,9]. Thus, optical integrated circuits can be fabricated using a laser beam that induces local structural changes in a fglass [15–18]. Photopolymerization and photodamage by highly focused laser pulses can be used in microchemistry and stereolithography [6,19]. Moreover, the femtosecond laser

structuring of glass-forming materials can be used for the optical long-term storage of information [3,20–30], photonics [18], the fabrication of phase gratings [31], nanogratings [32], and quantum dots that can be used in various devices [33–35]. Notably, optical storage based on glass-forming materials has the potential to replace magnetic storage in the quest to provide high-speed, high-capacity, low-power, low-cost, highly secure, and long-term data storage [25–30]. However, for the development of these technologies, theoretical models of thermal processes occurring during spatially selective laser-induced structuring of glass-forming materials are required. For example, understanding the dynamics of the laser crystallization process of amorphous Ge films is important for transistor technology, photovoltaic devices, particle detectors, and photodetectors [36]. In fact, the laser crystallization method makes it possible to control the local temperature inside the material and avoid random nucleation [36]. The ability to form and control the dynamics of changes in the temperature distribution $T(t,r)$ inside the material is very important for the opportunity to control the morphology of the laser-written structure inside the glass matrix [22,32,37]. Thus, a deep understanding of the thermal processes inside glass-forming materials under fast laser thermal perturbations is required. In this paper, we focus on local temperature changes, especially the local cooling rate $-\mathcal{R}(t,r)$ in the focal region during laser-writing processes in a glass matrix (where $\mathcal{R}(t,r)$ is the rate of temperature change).

Femtosecond laser-induced microstructuring is a promising tool because the structures induced by femtosecond laser pulses can be even smaller than the optical diffraction limit of the beam focusing since multiphoton absorption is most effective in the central part of the beam's focus. Typically, in laser-induced microstructuring, the laser pulse duration $\tau_{laser}$ is on the order of 100 fs, and the radiation wavelength is about 1 μm. When a transparent material is irradiated with a powerful femtosecond laser pulse focused into a micro-sized focusing zone, multiphoton absorption of the laser radiation energy via electrons of the irradiated material occurs. The energy of the electrons in the laser focus increases sharply during $\tau_{laser}$ due to electron–photon interaction. The electrons then transfer their energy to the lattice through electron–phonon interaction. The thermalization process takes approximately tens or hundreds of picoseconds [6,10,38,39]. Consequently, the duration of heating pulses $\tau_p$ that heat the material can be on the order of tens to hundreds of picoseconds. Thus, the material can be locally heated above the glass transition temperature $T_g$ and melting temperature $T_m$. Heat then spreads through the material, and the hot focal zone is rapidly cooled below $T_g$.

In this article, we will consider laser pulses of moderate energy $E_p$ (in the range 10–100 nJ), which is sufficient to locally heat the material well above the glass transition temperature $T_g$ but not sufficient to destroy the material with the formation of voids. Thus, the intensity of the laser pulses considered in this article is below the material damage threshold, which, for example, is about $10^{16}$ W/cm$^2$ for silica glass [40]. We focus on micrometer-scale local thermal perturbations caused by laser pulses near or below the threshold ionization intensity of a material. The threshold ionization intensity of most transparent solids ranges from $10^{13}$ to $10^{14}$ W/cm$^2$ for light with a wavelength of about 1 μm [41,42]. For example, the threshold intensity in silica glass and similar glass-forming materials is about $10^{13}$ W/cm$^2$ for light pulses with a wavelength of about 1 μm and a duration of 100 fs [41].

Thus, we will consider thermal processes in dielectric glasses below the threshold intensity of ionization of the material and in the time interval after electron–phonon thermalization, when there is no electron-hole plasma. Thus, we are interested in changes in the lattice temperature on a time scale outside the time interval when a multi-temperature model is usually considered [12–14]. In fact, we are interested in changes in lattice temperature that affect the structure of the material at moderate laser pulse energies and time $t$ exceeding the electron–phonon thermalization time.

The initial structure of glass-forming materials is not completely restored after the rapid heating–cooling cycle, which leads to a change in the local properties of the material in rapidly cooled areas. The local structure, specific volume, density, Rayleigh scattering

loss, and refractive index of glass-forming materials significantly depend on the local cooling rate [19,43,44]; see Figure 1.

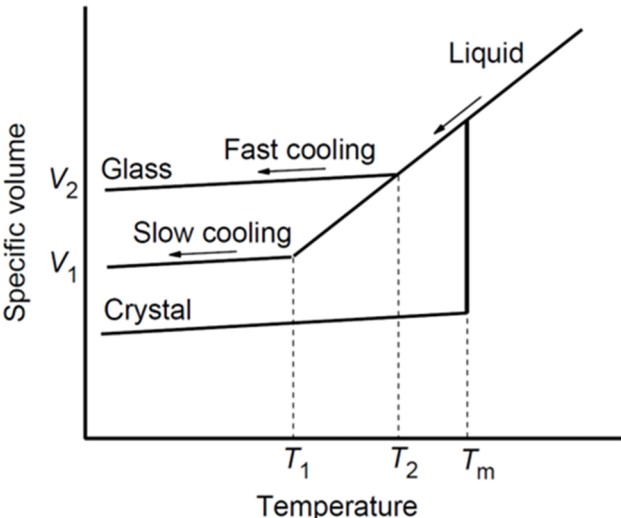

**Figure 1.** Schematic diagram of the specific volume $V$ as a function of temperature $T$ for glass-forming materials at different cooling rates of $-\mathcal{R}_1$ and $-\mathcal{R}_2$. Liquid structures freeze to a glassy state with $V_2(\mathcal{R}_2) > V_1(\mathcal{R}_1)$ at $|\mathcal{R}_2| > |\mathcal{R}_1|$.

Thus, we focus on the dynamics of the temperature distribution $T(t, r)$ and study the local cooling rate $-\mathcal{R}(t, r)$ near the focal region during laser-induced microstructuring. Measurements of the temperature distribution at the focus of the laser beam can be carried out using micro-Raman spectroscopy [45–47]. However, the time resolution of Raman spectroscopy (with a measuring pulse duration of about 10 ns and a repetition rate of 1 kHz) is not sufficient to detect ultrafast changes in the local temperature $T(t, r)$. We found that the local cooling rate $-\mathcal{R}(t, r)$ after the end of the heating pulse can reach more than $10^{11}$ K/s in a thin layer around the focus of the laser beam. In fact, resolving such ultrafast temperature changes at the periphery of the hot focal zone requires a temporal and spatial resolution of at least about 0.1 ns and 10 nm, respectively, since the cooling rate $-\mathcal{R}(t, r)$ reaches about 600 GK/s in a narrow layer around the hot zone at the laser focus (see below). Thus, the maximum change in the glass structure occurs at the periphery of the focal region (where the material undergoes ultrafast quenching). In fact, the laser microstructuring of glasses makes it possible to form micron-sized domains that are close to spherical or ring-shaped, with a modified glass structure at the periphery of the domains [19,33,35,39,45,48–52]. For example, the local modification of silver-doped phosphate glasses with laser pulse energies $E_p$ in the range of 10–100 nJ led to the formation of micron-sized ring-shaped domains due to the aggregation of silver nanoclusters at the periphery of the domains [48]. Laser pulse energy $E_p$ and pulse repetition rate play an important role in the microstructuring of glass-forming materials. In fact, laser pulses with a sufficient repetition rate can provide cumulative heating near the focus of the laser beam. In this article, we will focus on the effect of single laser pulses (the effect of repetition rate will be discussed in a separate article).

The temperature change associated with the heating pulse is on the order of $\Delta T = E_p / C_{loc}$, where $C_{loc} = \rho c_p V_0$ and $V_0$ are the heat capacity and volume of the heating zone, and $\rho$ and $c_p$ are the density and specific heat capacity, respectively, of the material. The thermal effect caused by elastic deformations is negligible compared to $\Delta T$, at least at a moderate $E_p$, in the range of, say, 10–100 nJ for a focal region with a radius of about 1 μm. Indeed, the thermoelastic pressure in the hot zone is less than the maximum pressure response $p_{max} = K_B \alpha_V \Delta T$ to the temperature change $\Delta T$ at constant volume, where $K_B$ and $\alpha_V$ are the bulk modulus and the volumetric thermal expansion coefficient. However, the volume of the hot zone is not

rigidly fixed by the material surrounding this zone. In fact, a pressure wave is created around the hot zone [53,54]. In this case, the thermoelastic pressure is even less than $p_{max}$. Thus, the elastic energy associated with the thermoelastic pressure in the hot zone of volume $V_0$ is less than $E_{TE} = \frac{1}{2}K_B(\alpha_V \Delta T)^2 V_0$. Therefore, the relative effect associated with thermoelastic deformations does not exceed $E_{TE}/E_p = \frac{1}{2}K_B(\alpha_V)^2 \Delta T / \rho c_p$. For example, for sodium-lime-silicate glasses with the composition $Na_2O \cdot 2CaO \cdot 3SiO_2$, $E_{TE}/E_p$ is about 1% at $\Delta T = 2000$ K, $K_B = 55$ GPa, $\rho = 2.8$ g/cm$^3$, $c_p = 1.14$ J/gK, $\alpha_V = 3\alpha$, and linear thermal expansion $\alpha = 7.7 \cdot 10^{-6}$ 1/K [55–59]. This ratio is even lower for borosilicate glasses with a low coefficient of thermal expansion [60]. For example, for borosilicate glasses such as Pyrex, the $E_{TE}/E_p$ ratio is about 0.1% at $\Delta T = 2000$ K, $K_B = 33$ Gpa, $\rho = 2.23$ g/cm$^3$, $c_p = 1.1$ J/gK, and $\alpha = 3.3 \cdot 10^{-6}$ 1/K [60–63]. However, $E_{TE}$ becomes comparable with $E_p$ at laser pulse energies approximately two orders of magnitude higher than those considered in this work. We are interested in local temperature changes in a micron-sized hot zone at $t > \tau_p$, when the pressure waves created around the hot zone are already at a distance of more than 1–2 μm from the hot zone. However, at the same time, the front of the temperature change extends only about ten nanometers from the periphery of the hot zone. Thus, we consider the change in local temperature after the local pressure in the hot zone has almost stabilized. For example, in silicate glasses at $t = 0.5$ ns, the distance $tV_L$ is about 3 μm at the longitudinal speed of sound $V_L \approx 6 \cdot 10^3$ m/s [58] and $\sqrt{tD_0} \approx$ 10 nm at the thermal diffusion coefficient $D_0$ of about $3 \cdot 10^{-7}$ m$^2$/s [56,62,64].

It Is noteworthy that an important property of glass-forming materials is the long-term relaxation of the specific heat capacity with rapid changes in temperature. Thus, the heat capacity of glass-forming materials must be considered as a function of time $c_{dyn}(t)$ [65–67]. In fact, if a glass-forming material is heated to a liquid state with heat capacity $c_L$ from a solid glassy state with heat capacity $c_S$, then the heat capacity of the material does not change immediately with the change in temperature but slowly relaxes from $c_S$ to $c_L$; see below. The long-term relaxation of the dynamic heat capacity $c_{dyn}(t)$ of glasses is due to the slow exchange of energy between different degrees of freedom in glasses. Thus, the thermal response of glass-forming materials to a thermal perturbation at time $t$ depends on the temperature at earlier times. The effect of the long-term relaxation of dynamic heat capacity significantly affects the dynamics of local temperature changes $T(t,r)$ [68,69]. In turn, the rate of change in local temperature $T(t,r)$ can significantly affect the microstructuring of glass-forming materials [19,37,43,44,70]. An experimental study of the dynamics of the temperature distribution $T(t,r)$ inside a glass matrix under the action of fast laser thermal perturbations is very difficult. Therefore, it is necessary to develop theoretical models of thermal processes occurring during the laser-induced structuring of glasses. In this paper, we will focus on modeling such thermal processes, taking into account the relaxation effect of the dynamic heat capacity $c_{dyn}(t)$. The dynamic behavior of glass-forming materials under fast local thermal perturbations can be described using the integro-differential heat equation with "memory" [68,69]. This equation has an analytical solution, at least in spherical, cylindrical, and planar geometries [68,69,71]. This work aims to determine the dynamics of local temperature changes, especially the local rate of the temperature change $\mathcal{R}(t,r)$ associated with laser-induced thermal excitations during laser processing of glasses. An analytical method for determining $T(t,r)$ and $\mathcal{R}(t,r)$ has been developed. The knowledge obtained can be useful for various technologies related to laser-induced microstructuring.

In the first part of the article, the heat equation with dynamic heat capacity $c_{dyn}(t)$ is considered. An analytical solution to this equation in general form was presented in our previous work [69], and then this method was applied to describe local thermal disturbances in supercooled glass-forming liquids and polymers during the nucleation of the crystalline phase [70]. We are currently using this method to study the rate of temperature change $\mathcal{R}(t,r)$ inside glasses under local fast laser excitations during laser-induced microstructuring. An analysis of the dynamic heat capacity $c_{dyn}(t)$ of glass-forming materials was conducted, and an analytical solution of the heat equation with dynamic heat capacity for a spherically symmetric problem was constructed. Then, the temperature distribution $T(t,r)$ was calculated for glasses with dynamic heat capacity $c_{dyn}(t)$ under local fast laser excitations. Examples with borosilicate and sodium-lime-silicate glasses

are considered at different heating pulses and dynamic heat capacity parameters. Finally, the local distribution of the cooling rate $-\mathcal{R}(t,r)$ and its influence on the microstructuring processes of glasses is discussed.

## 2. Heat Equation with Dynamic Heat Capacity

Since the pioneering work of Birge and Nagel [65], the dynamic heat capacity $c_{dyn}(t)$ of glass-forming materials as a function of frequency (or time $t$) has been intensively studied. For such studies, heat capacity spectroscopy can be used [66,67]. The time dispersion of the dynamic heat capacity $c_{dyn}(t)$ can be described within the framework of linear response theory [72], which is similar to the time dispersion of the dielectric constant [73]. Then, heat transfer in the glass-forming material can be described using an integro-differential heat equation [68,69]. In the case of spherical geometry and zero initial conditions, this heat equation can be represented in the following form [68,69]:

$$\frac{\partial}{\partial t}\int_0^t \rho c_{dyn}(t-\tau)T\prime(\tau,r)d\tau - \lambda\Delta T(t,r) = \Phi(t,r), \tag{1}$$

where $\Phi(t,r)$ is the volumetric heat flux density, $T'(t,r) = \frac{\partial}{\partial t}T(t,r)$, $\Delta$ is the Laplacian, and $\lambda$ and $\rho$ are the thermal conductivity and density of the material. Equation (1) has an analytical solution for homogeneous boundary conditions; see below.

### 2.1. Dynamic Heat Capacity of Glass-Forming Materials

The dynamic heat capacity $c_{dyn}(t)$ as a function of time can be represented as a continuous sum of exponential decays, since $c_{dyn}(t)$ is a monotonically relaxing function of time [68,69,74]:

$$c_{dyn}(t) = c_0 - (c_0 - c_{in})\int_0^\infty H(\tau_0)exp(-t/\tau_0)d\tau_0, \tag{2}$$

where $c_{in}$ and $c_0$ are the initial and final (equilibrium) heat capacities. $c_{dyn}(t) \to c_{in}$ as $t \to 0$ and $c_{dyn}(t) \to c_0$ as $t \to \infty$. In fact, $c_{in} = c_S$ and $c_0 = c_L$ if the material is heated from a solid glassy state to a liquid one. The distribution function $H(\tau_0)$ of relaxation times $\tau_0$ can be found using broadband heat capacity spectroscopy [67]; for details, see [68,69,71]. The dynamic heat capacity $c_{dyn}(t)$ is usually described using the Kohlrausch–Williams–Watts stretched exponent $exp(-(t/\tau_K)^\beta)$ for $0 < \beta \le 1$, where the Kohlrausch relaxation time $\tau_K$ and constant $\beta$ characterize the relaxation time spectrum of the material [44,75,76]. In this case,

$$c_{dyn}(t) = c_0\left[1 - \varepsilon_0 exp\left(-(t/\tau_K)^\beta\right)\right], \tag{3}$$

where $\varepsilon_0 = (c_0 - c_{in})/c_0$. Usually, $\varepsilon_0$ is about 0.3–0.5 [67,77]. For example, if $\beta = 0.5$, then $H(\tau_0) = \frac{exp(-\tau_0/4\tau_K)}{\sqrt{4\pi\tau_K\tau_0}}$ [75,76]. Thus, $c_{dyn}(t)$ can be approximately represented by Equation (4) for a sufficiently large finite interval $[\tau_{min}, \tau_{max}]$:

$$c_{dyn}(t,\tau_K) \approx c_0\left[1 - \varepsilon_0\int_{\tau_{min}}^{\tau_{max}}\frac{exp(-\tau_0/4\tau_K)}{\sqrt{4\pi\tau_K\tau_0}}exp(-t/\tau_0)d\tau_0\right], \tag{4}$$

where $\tau_K$ is a function of temperature. The function $\tau_K(T)$ can be obtained from the Vogel–Fulcher–Tammann–Hesse (VFTH) relation measured using broadband heat capacity spectroscopy [67]:

$$f_{max} = f_0 exp[-B/(T-T_0)], \tag{5}$$

where $f_{max}$ is the frequency corresponding to the maximum value of the temperature dependence of the imaginary part of the dynamic heat capacity $c_{dyn}(\omega)$; $\omega = 2\pi f$ is the temperature modulation frequence; and $f_0$, $B$, and $T_0$ are the VFTH parameters. In fact, $\tau_K\omega$ is about 1. More precisely, $\tau_K$ for different $T$ can be obtained from the relation $\tau_K = \frac{0.737}{2\pi f_{max}}$ [68]. However, the shape of the distribution function $H(\tau_0)$ has insignificant effect on $T(t,r)$, since the effect due to the time dispersion of the dynamic heat capacity

reaches saturation with increasing $\tau_0$; see below. Thus, it is sufficient to consider the dynamic heat capacity with the Debye relaxation law (see Equation (6)) for sufficiently large $\tau_0$. Below, we will show that this result practically coincides with the result obtained after averaging over the distribution function $H(\tau_0)$. In the case of the Debye relaxation law, $c_{dyn}(t)$ is equal to

$$c_{dyn}(t) = c_0[1 - \varepsilon_0 exp(-t/\tau_0)]. \tag{6}$$

In this case, the integro-differential heat equation, Equation (1), is transformed into Equation (7), which can be solved analytically. After this, solutions for different $\tau_0$ can be averaged using the distribution function $H(\tau_0)$.

### 2.2. Analytical Solution of the Heat Equation with Dynamic Heat Capacity

Let us consider the temperature distribution $T(t, r)$ after a single laser pulse in a micron-sized focal zone. A relatively small change in the thermal conductivity of the glass matrix around the hot focal zone does not affect the dynamics of the temperature distribution in the hot focal zone. In fact, for the glass-forming materials under consideration, $\lambda$ varies insignificantly with temperature and is about 1 W/m$^2$K over a wide temperature range [64,78,79]. Thus, it is assumed that the thermal conductivity $\lambda$ of the glass matrix does not depend on temperature. However, the change in dynamic heat capacity $c_{dyn}(t)$ from $c_S$ to $c_L$ during heating is significant.

As a first step, we consider the Debye relaxation law; see Equation (6). Thus, for spherical geometry, from Equation (1), we obtain Equation (7):

$$\frac{\partial}{\partial t}T(t, r) - D_0 \Delta T(t, r) = \frac{\Phi(t, r)}{\rho c_0} + \varepsilon_0 \frac{\partial}{\partial t}\int_0^t exp\left(-\frac{t - \tau}{\tau_0}\right) T\prime(t, r)d\tau, \tag{7}$$

where $D_0 = \lambda/\rho c_0$ and the spherically symmetric heat source $\Phi(t, r) = \Phi(r)F(t)$ is distributed in a spherical volume of radius $r_0$. The decomposition of $\Phi(t, r)$ into the product $\Phi(r)F(t)$ is common for laser heating. Furthermore, after the end of the heating pulse, the temperature distribution $T(t, r)$ does not depend on the shape of $F(t)$; see below. In this article, we are interested in the dynamics of the temperature distribution $T(t, r)$ after the heating pulse, when the shape of the heating pulse does not matter.

Consider Equation (7) under the initial conditions $\Phi(t, r) = 0$ and $T(t, r) = T_{in}$ for $t \leq 0$, where $T_{in}$ is the initial temperature in the glass matrix. Let $T(t, r)$ be a bounded function and $T(t, R_0) = T_{in}$ for $R_0 \gg r_0$. Since we are considering a single heating pulse, it does not matter how large the parameter $R_0$ is, as long as $R_0 \gg r_0$. Indeed, the temperature change is localized in a hot zone about a few micrometers in size; see below. The solution to this boundary value problem can be represented in the form of the following series (for details, see Appendix A):

$$T(t, r) = T_{in} + \sum_{n=1} \psi_n(t) \frac{sin(\pi n r/R_0)}{r}, \tag{8}$$

where the functions $\psi_n(t)$ are determined by Equation (9).

$$\psi'_n(t) + \frac{\psi_n(t)}{\tau_n} = \frac{\Phi_n F(t)}{\rho c_0} + \varepsilon_0 \frac{\partial}{\partial t}\int_0^t exp\left(-\frac{t - \tau}{\tau_0}\right)\psi'_n(\tau)d\tau, \tag{9}$$

where $\Phi_n = \frac{2}{R_0}\int_0^{R_0} r\Phi(r)sin(\pi n r/R_0)dr$ and $\tau_n^{-1} = D_0(\pi n/R_0)^2$. The functions $\psi_n(t)$ satisfying Equation (9) are presented in Appendix A.

The spatial distribution $\Phi(r)$ of the heating power $\Phi(t, r)$ can be arbitrary. For example, we consider a heat source uniformly distributed in a volume of radius $r_0$ with volumetric density $\Phi_0$ (in W/m$^3$); then,

$$\Phi_n = 2R_0\Phi_0 \frac{sin(\pi n r_0/R_0) - (\pi n r_0/R_0) \cdot cos(\pi n r_0/R_0)}{(\pi n)^2}. \tag{10}$$

In fact, the distribution of the intensity of light radiation in the spot of a focused laser beam, depending on the distance $r$ from the center of the spot, can have a Gaussian bell-shaped character; see Figure 2. However, the intensity sufficient to induce nonlinear processes; for example, multiphoton absorption is located close to the center of the laser beam spot [5]. Thus, the model of a heat source uniformly distributed in a volume of radius $r_0$ with a sharp drop at the periphery is more realistic than the bell-shaped Gaussian distribution.

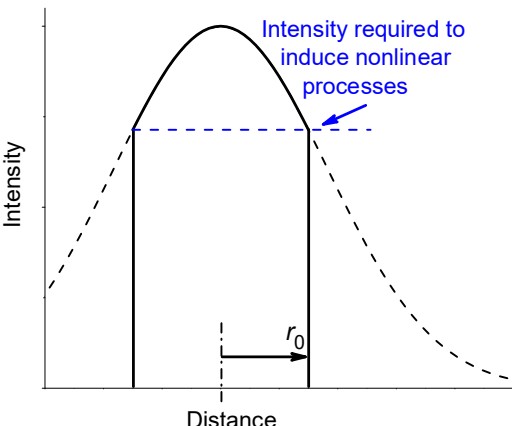

**Figure 2.** Schematic diagram of the distribution of the intensity of light radiation in a spot of a focused laser beam depending on the distance $r$ from the center of the spot (dashed line) and the intensity distribution sufficient to induce nonlinear processes such as multiphoton absorption in a spot with radius $r_0$ (solid line).

The shape of the change in heating power over time can be bell-shaped or more complex. However, this shape is not significant for the temperature distribution $T(t, r)$ at $t > \tau_p$; see below. For model calculations, we will use a heating pulse of a half-sinusoidal shape, which is slightly different from the bell-shaped one. Thus, $F(t) = sin(\pi t / \tau_p)$ for $0 \leq t \leq \tau_p$ and $F(t) = 0$ for $t > \tau_p$.

## 3. Temperature Distribution $T(t, r)$ in Glasses under Local Thermal Perturbations

For example, let us consider the two most common types of silicate glasses: sodium-lime-silicate glasses ($Na_2O$-$CaO$-$SiO_2$), as the most widely used of all industrial glasses, known for their low cost and availability, and borosilicate glasses ($B_2O_3$-$SiO_2$), which are widely used due to their low coefficient of thermal expansion and high resistance to chemical attack. Sodium-lime-silicate glass with the composition $Na_2O \cdot 2CaO \cdot 3SiO_2$ and borosilicate glass of the Pyrex type were chosen as model systems due to the availability of the necessary parameters [56,57,60–64]. The thermal parameters of these glasses are collected in Table 1.

**Table 1.** Thermal parameters of silicate glasses used for model calculations.

| Substance | Density in Solid State $\rho$ g/cm$^3$ | Specific Heat Capacity in the Solid State $c_p$ J/g·K | Volumetric Heat Capacity in Solid State $c_S$ J/m$^3$K | Volumetric Heat Capacity in Liquid State $c_L$ J/m$^3$K | Dynamic Heat Capacity Parameter $\varepsilon_0=$ $(c_L-c_S)/c_L$ | Thermal Conductivity $\lambda$ W/mK |
|---|---|---|---|---|---|---|
| Sodium-lime-silicate glass | 2.8 | 1.14 | $3.2 \cdot 10^6$ | $4.4 \cdot 10^6$ | 0.273 | 1 |
| Borosilicate glass | 2.23 | 1.1 | $2.45 \cdot 10^6$ | $3.8 \cdot 10^6$ | 0.355 | 1 |

### 3.1. Influence of the Shape and Duration of Heating Pulses on $T(t, r)$

Thus, for model calculations, we use a heating pulse with a power density $\Phi_0 sin(\pi t / \tau_p)$, acting during the time interval $\tau_p$ in a spherical region with a radius $r_0 = 1$ μm. Let us com-

pare the results for heating pulses with $F(t)$ with half-sinusoidal and rectangular shapes; see Figures 3 and 4. Let the initial temperature $T_{in} = 300$ K, $\tau_p = 300$ ps, $E_p = 15$ nJ, and the volume of the spherical heating zone $V_0 = \frac{4\pi}{3} r_0^3$. Thus, $\Phi_0$ is equal to $\frac{\pi}{2} E_p / \tau_p V_0$ and $E_p / \tau_p V_0$ for half-sinusoidal and rectangular heating pulses, respectively. In this case, the energy of these pulses is the same and equal to $E_p$. The temperature distribution $T(t, r)$ can be calculated according to Equations (8) and (9) at various $\tau_0$.

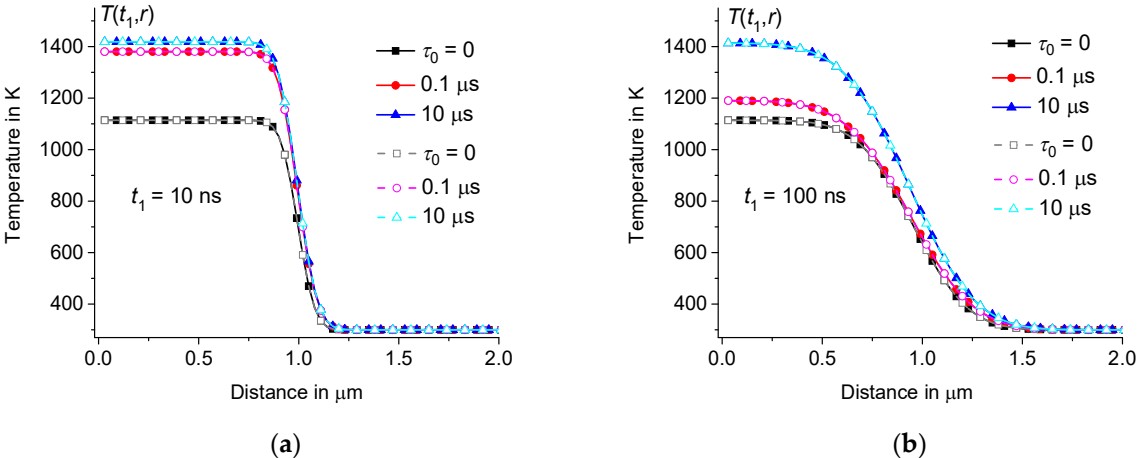

**Figure 3.** Temperature distribution $T(t_1, r)$ in the hot zone depending on the distance $r$ at $t_1 = 10$ ns (**a**) and $t_1 = 100$ ns (**b**) for heating pulses of half-sinusoidal shape (filled symbols) and rectangular shape (open symbols) in sodium-lime-silicate glass at $r_0 = 1$ μm, $\tau_p = 300$ ps, $E_p = 15$ nJ, and $T_{in} = 300$ K ($\tau_0 = 0$, 0.1 μs, and 10 μs—squares, circles, and triangles, respectively).

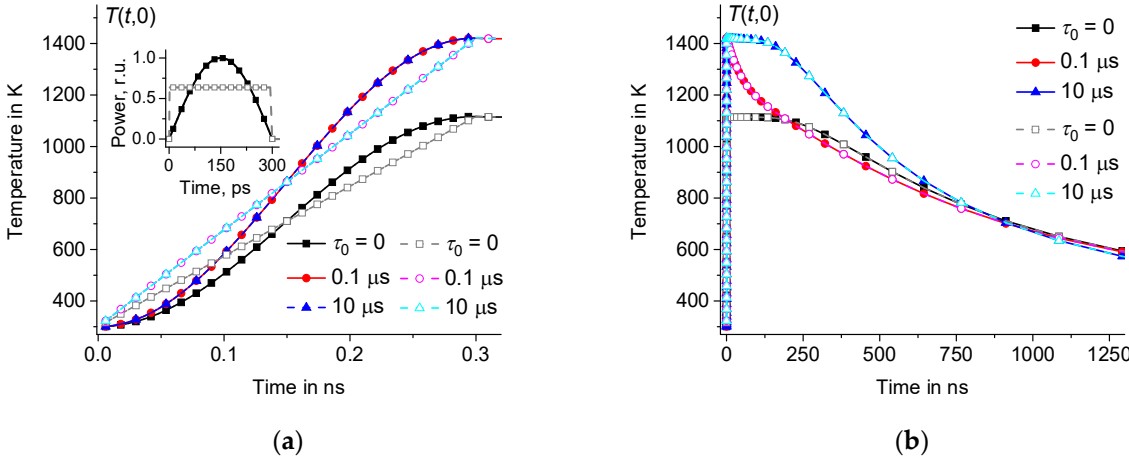

**Figure 4.** Time dependence of $T(t, 0)$ in the center of the hot zone during (**a**) and after (**b**) the heating pulses of half-sinusoidal shape (filled symbols) and rectangular shape (open symbols) at the same parameters as in Figure 3 ($\tau_0 = 0$, 0.1 μs, and 10 μs—squares, circles, and triangles, respectively). Inset (**a**) shows pulse power as a function of time.

For example, for sodium-lime-silicate glass (see Table 1), $T(t, r)$ at $\tau_0 = 0$, 0.1 μs, and 10 μs is represented in Figures 3 and 4. The temperature distribution $T(t, r)$ is calculated at $R_0 \gg r_0$ (for example, at $R_0 = 30$ μm). It follows from direct calculations that the result does not depend on $R_0$, at least on the time scale $t < R_0^2 / 4D_0$, where $R_0^2 / 4D_0$ for the glasses under consideration is about 3–4 ms. However, we are interested in changes in the temperature distribution $T(t, r)$ on a time scale of microseconds and less when the temperature changes are localized within a few micrometers; see Figure 3.

The influence of the time dispersion of the dynamic heat capacity $c_{dyn}(t)$ is significant; see Figures 3 and 4. The difference between equilibrium ($\tau_0 = 0$) and non-equilibrium

($\tau_0 \neq 0$) solutions $T(t, r)$ increases with the growth of $\tau_0$. The shape of the change in heating power over time is significant only during the time interval $(0, \tau_p)$; see Figure 4a. However, this shape is not significant at $t > \tau_p$; see Figures 3 and 4b.

The duration of the heating pulses is also not significant for the temperature distribution $T(t, r)$ at $t > \tau_p$; see Figure 5. For example, compare $T(t, r)$ for pulses with $\tau_p = 10$ ps and 300 ps and the same parameters as in Figure 3. The duration of heating pulses is significant only during the time interval $(0, \tau_p)$; see Figure 5a. However, this duration is not significant at $t > \tau_p$; see Figure 5b.

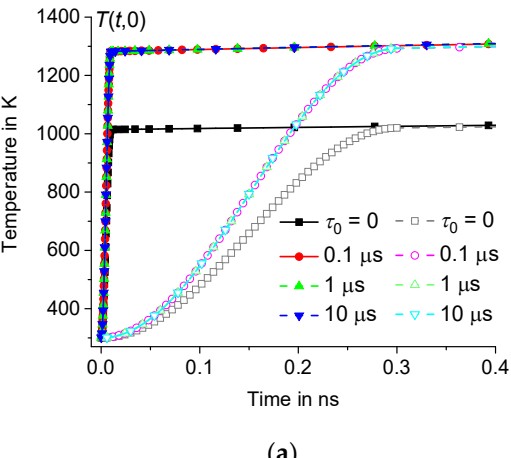

(**a**)

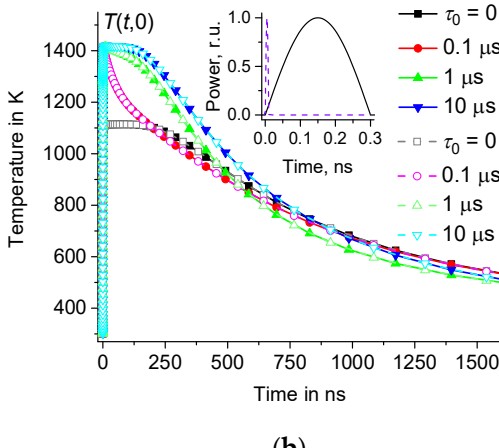

(**b**)

**Figure 5.** Time dependence of $T(t, 0)$ in the center of the hot zone during (**a**) and after (**b**) the heating pulse of half-sinusoidal shape at $\tau_p = 10$ ps and 300 ps, represented by filled and open symbols, respectively, at the same parameters as in Figure 3 ($\tau_0 = 0$, 0.1 μs, 1 μs, and 10 μs—squares, circles, triangles up, and triangles down, respectively). Inset (**b**) shows the pulse power as a function of time for short and long pulses, represented by dotted and solid lines, respectively.

Thus, the temperature $T(t, 0)$ at the center of the hot zone increases very quickly over the time interval $(0, \tau_p)$ from $T_{in}$ to the maximum value $T_{max}$ and then relaxes relatively slowly on a microsecond time scale; see Figures 4 and 5. The influence of the time dispersion of dynamic heat capacity is significant on time scales of several hundred nanoseconds or less. The solution associated with the dynamic heat capacity (at $\tau_0 \neq 0$) tends to the classical solution (corresponding to $\tau_0 = 0$) when $t$ reaches approximately 1 μs; see Figures 4b and 5b.

### 3.2. Comparison with the Fundamental Solution of the Classical Problem

It is often believed that the hot focal spot should spread and cool down according to the law that $r \sim (D_0 t)^{1/2}$ and $T \sim t^{-3/2}$, respectively [19,46,80,81]. This assumption is based on the fundamental solution of the Fourier heat equation, which can be represented by the following function [82]:

$$G\left(t, \vec{r}\right) = \theta(t) \exp\left(-\left|\vec{r}\right|^2 / 4D_0 t\right) / (4\pi D_0 t)^{3/2}, \tag{11}$$

where $\theta(t)$ is the Heaviside unit step function and $r$ is the distance from the instantaneous point heat source. However, for a non-point heat source, the approximation of $T(t, r)$ by this function is not satisfactory. The real dependence $T(t, r)$ differs significantly from Equation (11) [45]. As expected, the temperature perturbation $\delta T(t) = T(t, 0) - T_{in}$ in the center of the heating zone relaxes in a time of the order of $r_0^2 / D_0$. However, $\delta T(t)$ remains near the maximum value $\delta T(\tau_p)$ at $t < 0.1$ μs (see Figure 6), and only then does $\delta T(t)$ relax over a time period on the order of $r_0^2 / D_0$. In contrast to the estimate based on the fundamental solution of the Fourier heat equation, the temperature perturbation $\delta T(t)$ relaxes approximately as $T \sim t^{-1}$ and not as $T \sim t^{-3/2}$; see Figure 6. In order to obtain

the correct temperature distribution $T(t,r)$ using the fundamental solution, it is necessary to integrate the thermal response over the time interval $(0, \tau_p)$ and the volume of the heating zone. In fact, the solution of the generalized Cauchy problem for the classical heat equation with the heat source function $\Phi(t,r)$ under zero initial conditions is expressed by the classical Poisson formula (see Equation (12)), where $G\left(t, \overrightarrow{r}\right)$ is the fundamental solution of the classical heat equation [82]. This fundamental solution is represented by Equation (11).

Thus, for the heat source $\Phi(t,r)$ considered above, distributed in a volume with radius $r_0$ and acting on the time interval $(0, \tau_p)$, we obtain the temperature distribution $T_{FS}(t,r)$; see Equation (12).

$$T_{FS}(t,r) = \int_0^{r_0} \int_0^{\tau_p} \int_0^{\pi} \frac{\Phi_0 F(\tau)}{\rho c_0} G\left(t - \tau, \overrightarrow{r} - \overrightarrow{\xi}\right) 2\pi sin(\theta)\xi^2 d\xi d\tau d\theta, \qquad (12)$$

where $\left(\overrightarrow{r} - \overrightarrow{\xi}\right)^2 = r^2 + \xi^2 - 2r\xi cos(\theta)$. Integration in Equation (12) was carried out using standard mathematical engineering software Mathcad 15.0. Integration over the variables $\xi$ and $\theta$ was carried out directly using Mathcad software. However, to avoid singularity, time integration was replaced with a discrete sum so that the difference between $t$ and $\tau$ was at least 0.3 ps. In fact, a sum of 30 terms was enough to obtain good accuracy with a sum normalization factor of 0.969 required to normalize the total energy of the heating pulse. In addition, the integration over all variables can be carried out directly using Mathcad software for $t > 2\tau_p$, which is far from the singularity of the fundamental solution.

As expected, the solution $T_{FS}(t,r)$ completely coincides with the temperature distribution $T(t,r)$, calculated using Equations (8) and (9) at $\tau_0 = 0$; see Figure 6. In contrast, the simplified estimate $\mathcal{T}(t,r) = N_{cor} \frac{\Phi_0}{\rho c_0} G\left(t, \overrightarrow{r}\right) \tau_p V_0$ is far from the correct $T_{FS}(t,r)$ even for any correction factor $N_{cor}$, see Figure 6.

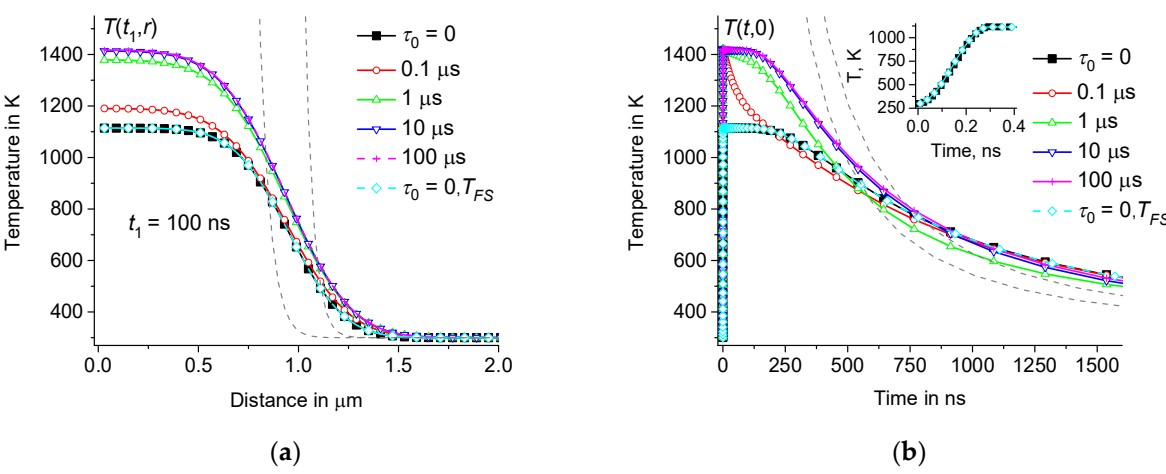

**(a)**                                      **(b)**

**Figure 6.** $T(t_1,r)$ depending on the distance $r$ at $t_1 = 100$ ns (**a**) and time dependence of $T(t,0)$ in the center of the hot zone (**b**) at the same parameters as in Figure 3 ($\tau_0 = 0$, 0.1 µs, 1 µs, 10 µs, and 100 µs—squares, circles, triangles up, triangles down, and crosses, respectively, as well as $T_{FS}(t,r)$ for $\tau_0 = 0$—diamonds). Dotted lines show $\mathcal{T}(t_1,r)$ at $N_{cor} = 1$ and 100 (**a**), as well as $\mathcal{T}(t,0)$ at $N_{cor} = 1/150$ and 1/200 (**b**). Inset (**b**) shows the initial fragments of $T_{FS}(t,0)$ and $T(t,0)$ at $\tau_0 = 0$.

Thus, the correct calculation of the temperature distribution $T(t,r)$ using the fundamental solution of the classical heat equation (without time dispersion) gives the same result as the calculation using Equations (8) and (9) at $\tau_0 = 0$. However, using Equations (8) and (9), it is possible to calculate the temperature distribution $T(t,r)$ for materials with dynamic heat capacity (at $\tau_0 \neq 0$). It turns out that $T(t,r)$ obtained at $\tau_0 \neq 0$ increases with $\tau_0$ and tends to saturation at $\tau_0$ about 10 µs; see Figure 6. For this reason, the shape of the distribution function

$H(\tau_0)$ has an insignificant effect on $T(t, r)$. Next, we consider the influence of this distribution on $T(t, r)$; see Figures 7 and 8.

### 3.3. Dependence of $T(t, r)$ on the Distribution of Relaxation Times

The influence of the time dispersion of the dynamic heat capacity is most pronounced at the beginning of the heating process on a nanosecond time scale; see Figure 6b. This effect is significant already at $\tau_0$ about 0.1 μs, increases with increasing $\tau_0$, and it reaches saturation at $\tau_0$ above $r_0^2/D_0$, where $r_0^2/D_0$ is about 4 μs at $r_0 = 1$ μm and $D_0 = 2.6 \cdot 10^{-7}$ m²/s for borosilicate glass. In fact, in glasses, the relaxation times $\tau_0$ are distributed over a wide range. The parameters of the VFTH relationship (see Equation (5)) can be obtained from measurements of glass transition processes depending on the cooling rate. For example, we use these parameters for borosilicate glass [63], sodium-lime-magnesium-silicate glass [83], and sodium-silicate glass [84]. The relaxation parameters of these silicate glasses used for model calculations are collected in Table 2.

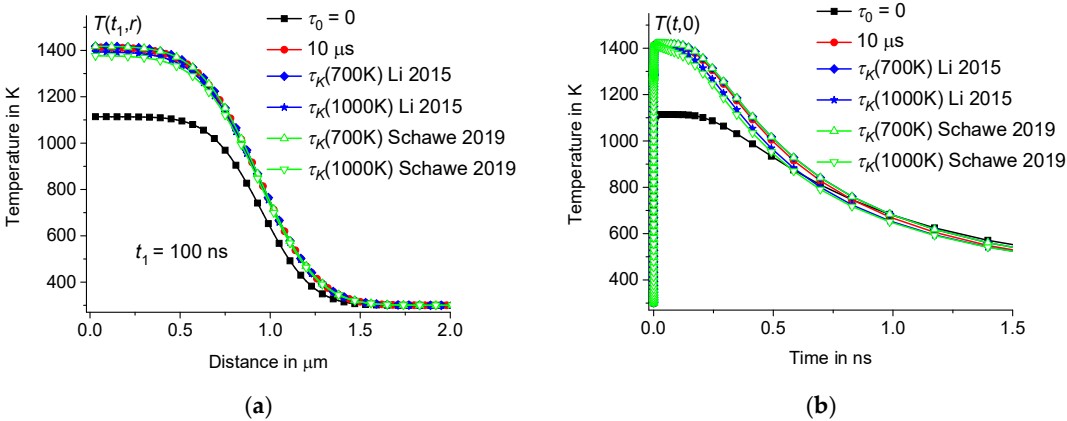

(a)

(b)

**Figure 7.** $T(t_1, r)$ depending on the distance $r$ at $t_1 = 100$ ns (a) and time dependence $T(t, 0)$ in the center of the hot zone (b) at the same parameters as in Figure 3 ($\tau_0 = 0$ and 10 μs—squares and circles). The temperature distribution $T_{AV}(t, r)$ is averaged using the distribution function $H(\tau_K(T_{int}))$ obtained from [84] at $T_{int} = 700$ K and 1000 K—diamonds and stars, as well as from [83] at $T_{int} = 700$ K and 1000 K, which are represented by triangles facing up and triangles facing down, respectively.

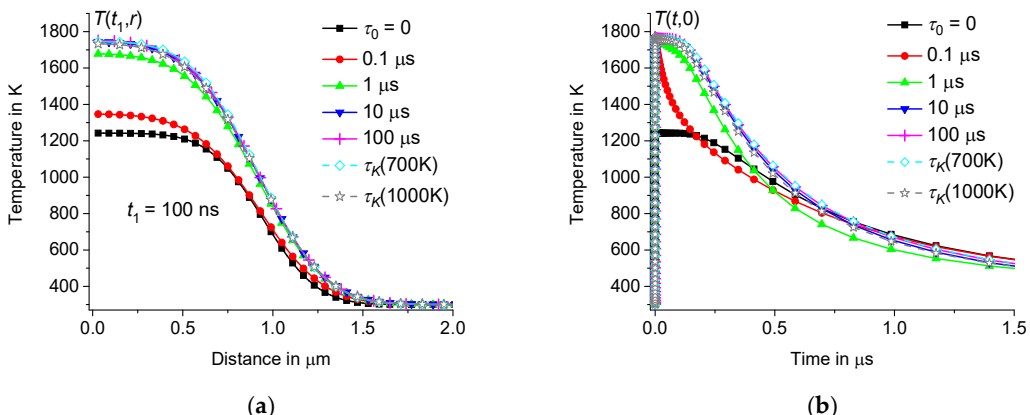

(a)

(b)

**Figure 8.** $T(t_1, r)$ depending on the distance $r$ at $t_1 = 100$ ns (a) and time dependence $T(t, 0)$ in the center of the hot zone (b) for borosilicate glass at $r_0 = 1$ μm, $\tau_p = 10$ ps, and $E_p = 15$ nJ $T_{in} = 300$ K ($\tau_0 = 0$, 0.1 μs, 1 μs, 10 μs, and 100 μs—squares, circles, triangles up, triangles down, and crosses). The solution $T_{AV}(t, r)$ averaged using the distribution function $H(\tau_K(T_{int}))$ obtained from [63] at $T_{int} = 700$ K and 1000 K, which are represented by diamonds and stars, respectively.

**Table 2.** Relaxation parameters of silicate glasses used for model calculations.

| Substance | $f_0$ Hz | $B$ K | $T_0$ K |
|---|---|---|---|
| Borosilicate glass | $5 \cdot 10^{17}$ | $1.9 \cdot 10^4$ | 392 |
| Sodium-lime-magnesium-silicate glass | $1 \cdot 10^{14}$ | $1.1 \cdot 10^4$ | 480 |
| Sodium-silicate glass | $1.24 \cdot 10^{14}$ | $1.29 \cdot 10^4$ | 407 |

The dynamic heat capacity $c_{dyn}(t)$ can be modeled using Equation (4), where $\tau_K(T) = \frac{0.737}{2\pi f_{max}}$ can be obtained from $f_{max}$ (see Equation (5)) using the relaxation parameters; see Table 2. Thus, we obtain the temperature distribution $T_{AV}(t,r)$ averaged using the distribution function $H(\tau_K(T_{int}))$, where $\tau_K(T_{int})$ is obtained at several intermediate temperatures between $T_{in}$ and $T_{max}$ (where $T_{max}$ is the maximum value of the temperature $T(t,0)$ at the center of the hot zone). For example, set $T_{int} = 700$ K and 1000 K and consider the relaxation parameters for borosilicate glass [63], sodium-lime-magnesium-silicate glass [83], and sodium-silicate glass [84], see Table 2. The results are presented in Figures 7 and 8. The difference between $T(t,r)$, obtained for sufficiently large $\tau_0 \geq 10$ µs, and $T_{AV}(t,r)$, obtained for $\tau_K(700$ K$)$ and $\tau_K(1000$ K$)$, is insignificant; see Figures 7 and 8. Since the influence of the time dispersion of the dynamic heat capacity reaches saturation at $\tau_0$ about 10 µs, the shape of the distribution function $H(\tau_0)$ has little effect on the temperature distribution. Thus, the influence of the time dispersion of the dynamic heat capacity on the temperature distribution $T(t,r)$ can be calculated for a fixed, sufficiently large $\tau_0$.

The influence of time dispersion of dynamic heat capacity is significant in both borosilicate and sodium-silicate glasses. Now, let us consider the influence of the size of the heating zone $r_0$ on the temperature distribution $T(t,r)$.

*3.4. Dependence of $T(t,r)$ on the Size of the Heating Zone*

For example, let us compare the temperature distributions $T(t,r)$ at $r_0 = 1$ and $r_0 = 2$ µm for borosilicate glass; see Table 1. To obtain the same thermal response value, let us set $E_p = 120$ nJ at $r_0 = 2$ µm, increasing the energy in proportion to the volume of the heating zone. Thus, in both cases, we obtain the same amplitudes of the temperature response to heating pulses; see Figures 8 and 9. As expected, the temperature $T(t,0)$ in the center of the heating zone relaxes in a time period of the order of $r_0^2/D_0$, which for borosilicate glass is about 15 µs at $r_0 = 2$ µm and $D_0 = 2.6 \cdot 10^{-7}$ m$^2$/s. For comparison, $T(t,0)$ relaxes four times faster at $r_0 = 1$ µm than at $r_0 = 2$ µm; see Figures 8b and 9b. As the size of the heating zone increases, the saturation of the influence of the time dispersion of the dynamic heat capacity on $T(t,r)$ shifts towards larger values of $\tau_0$. For example, compare $T(t,r)$ at $\tau_0 = 1$ µs for $r_0 = 1$ and $r_0 = 2$ µm; see Figures 8 and 9.

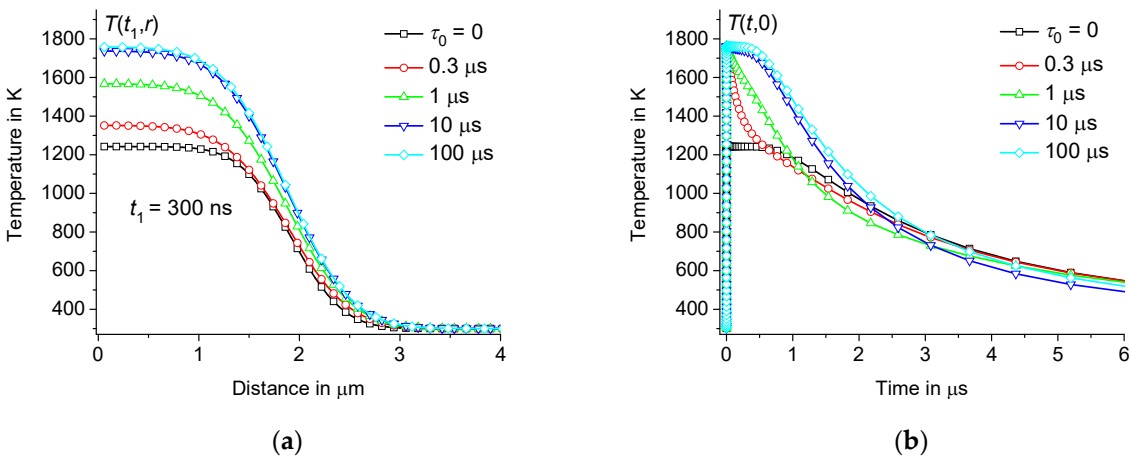

**Figure 9.** $T(t_1, r)$ depending on the distance $r$ at $t_1 = 300$ ns (**a**) and time dependence of $T(t, 0)$ in the center of the hot zone (**b**) for borosilicate glass at $r_0 = 2$ μm, $\tau_p = 300$ ps, $E_p = 120$ nJ, $T_{in} = 300$ K ($\tau_0 = 0$, 0.3 μs, 1 μs, 10 μs, and 100 μs—squares, circles, triangles facing up, triangles facing down, and diamonds).

## 4. Cooling Rate Distribution $-\mathcal{R}(t, r)$ and Its Influence on Microstructuring Processes of Glasses

Now consider the spatial distribution of the rate of temperature change $\mathcal{R}(t, r)$. Due to thermal expansion, the size of the heating zone changes with temperature by about 10 nm or less. We neglect these changes with respect to $r_0$. Let us consider the rate of temperature change $\mathcal{R}(t, r) = \frac{\partial T(t, r)}{\partial t}$ due to thermal diffusion. This temperature change affects the physical properties of the material. We will focus on very fast (about $10^9$ K/s or more) local temperature changes that have the greatest impact on the microstructuring process and will study the effect associated with the time dispersion of the dynamic heat capacity. It is worth noting that this effect is significant even well above the glass transition temperature $T_g$.

For example, the rate of temperature change $\mathcal{R}(t, r)$ for borosilicate glass at $r_0 = 2$ μm, $\tau_p = 300$ ps, $E_p = 120$ nJ and various $\tau_0$ values is presented in Figures 10 and 11. Note that the rate of temperature change $\mathcal{R}(t, r)$ is greatest near the periphery of the hot zone; see Figure 10a. The cooling rate $-\mathcal{R}(t, r)$ is about 600 GK/s at $r_1 = 1.98$ μm and $t_1 = 0.36$ ns; see Figure 11a. However, in the center of the heating zone, the cooling rate $-\mathcal{R}(t, 0)$ does not exceed 0.6 GK/s; i.e., $-\mathcal{R}(t, 0)$ is three orders of magnitude less than the cooling rate at the periphery—see Figure 11. Therefore, the local structure of glass in the center of the hot zone should differ significantly from the structure at the periphery of the heating zone after laser modification. This conclusion is consistent with experiments [19,33,35,39,45,48–52]. However, the effect associated with dynamic heat capacity (at $\tau_0 \neq 0$) is significant both in the center and at the periphery of the heating zone. Indeed, the maximum cooling rate is approximately 2.5 times greater at $\tau_0 \neq 0$ than at $\tau_0 = 0$ both at the periphery and in the center of the heating zone; see Figure 11.

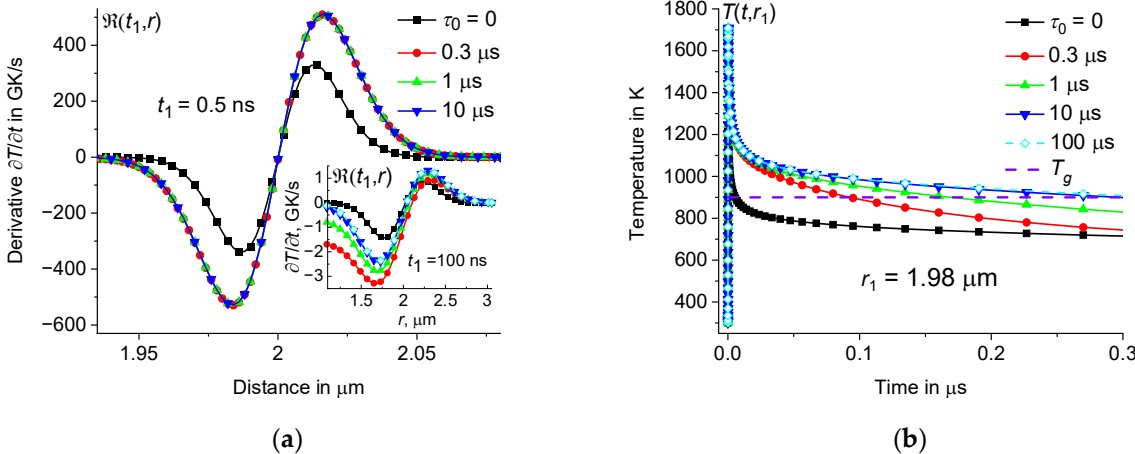

**Figure 10.** Rate of temperature change $\mathcal{R}(t_1, r)$ depending on the distance $r$ at the periphery of the hot zone at $t_1 = 0.5$ ns (**a**) and time dependence of $T(t, r_1)$ at $r_1 = 1.98$ µm (**b**) at the same parameters as in Figure 9 ($\tau_0 = 0$, 0.3 µs, 1 µs, 10 µs, and 100 µs—squares, circles, triangles up, triangles down, and diamonds). Inset (**a**) shows $\mathcal{R}(t_1, r)$ as a function of distance $r$ at $t_1 = 100$ ns for different $\tau_0$.

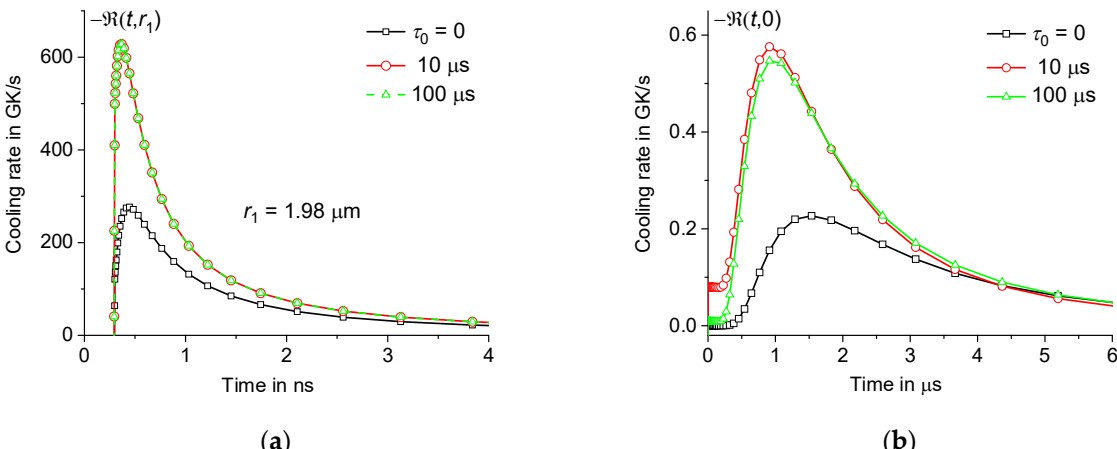

**Figure 11.** Time dependence of the cooling rate $-\mathcal{R}(t, r_1)$ at $r_1 = 1.98$ µm (**a**) and $r_1 = 0$ (**b**) at the same parameters as in Figures 9 and 10 ($\tau_0 = 0$, 10 µs, and 100 µs—squares, circles, and triangles).

Let us consider the cooling rate $-\mathcal{R}(t, r)$ in a thin shell in the region from $0.95r_0$ to $0.99r_0$. As we have established for borosilicate and sodium-lime-silicate glasses, ultrafast cooling of the material occurs in this region at a rate of more than $10^{11}$ K/s; see Figures 10–13. It is noteworthy that at the beginning of the cooling process (during the first few nanoseconds), the rate of temperature change $\mathcal{R}(t, r)$ is the same for small and large relaxation times $\tau_0$; see Figures 10a and 12a. This means that at the beginning of the cooling process at the periphery of the hot zone, the shape of the distribution function $H(\tau_0)$ does not affect the cooling rate $-\mathcal{R}(t, r)$. Therefore, in this case, even at very small relaxation times $\tau_0$, the influence of the time dispersion of the dynamic heat capacity is significant.

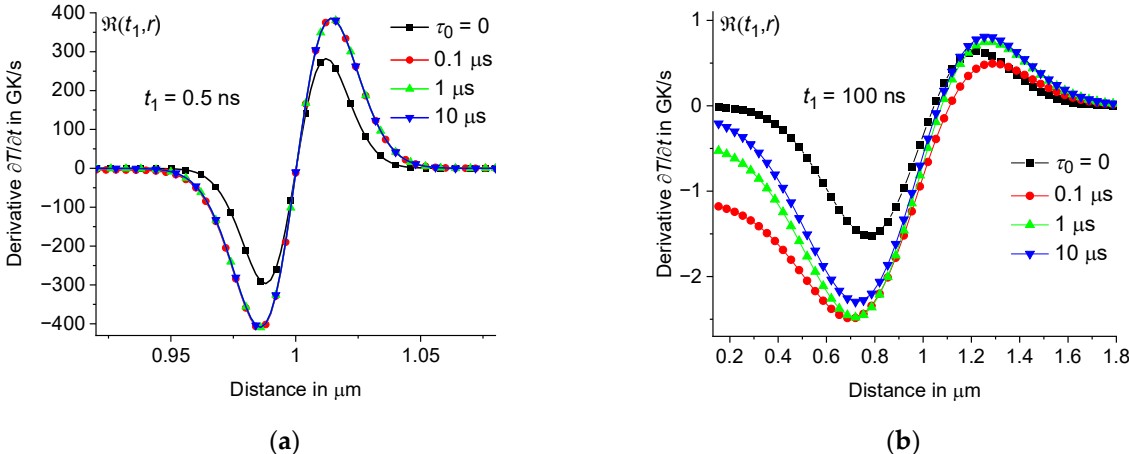

(**a**)                    (**b**)

**Figure 12.** $\mathcal{R}(t_1, r)$ depending on the distance $r$ at the periphery of the hot zone at $t_1 = 0.5$ ns (**a**) and $t_1 = 100$ ns (**b**) for sodium-lime-silicate glass at the same parameters as in Figure 3 ($\tau_0 = 0$ µs, 0.1 µs, 1 µs, and 10 µs—squares, circles, triangles up, and triangles down).

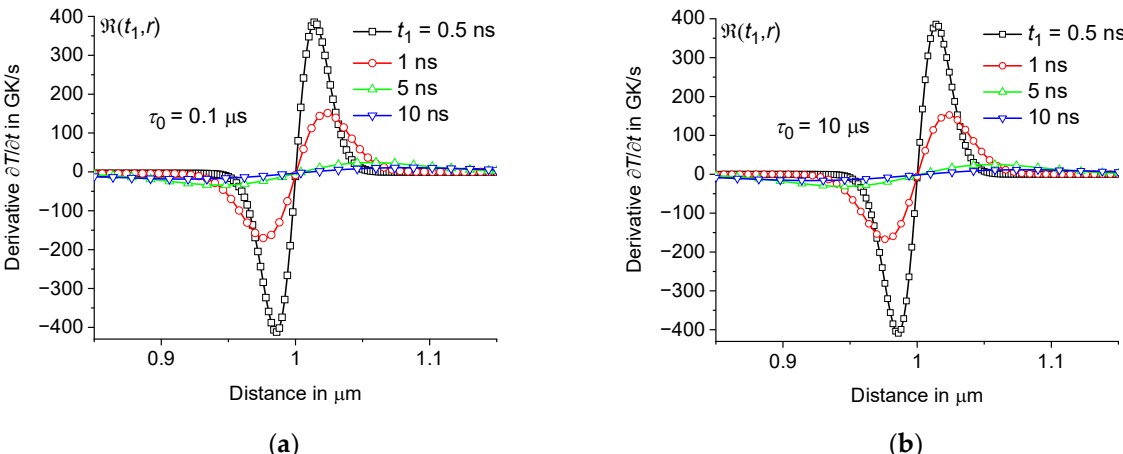

(**a**)                    (**b**)

**Figure 13.** $\mathcal{R}(t_1, r)$ depending on the distance $r$ at the periphery of the hot zone at $\tau_0 = 0.1$ µs (**a**) and $\tau_0 = 10$ µs (**b**) at the same parameters as in Figure 3 ($t_1 = 0.5$ ns, 1 ns, 5 ns, and 10 ns—squares, circles, triangles up, and triangles down).

The cooling rate $-\mathcal{R}(t_1, r)$ reaches a maximum at $r_{max}$ for a given $t_1$; see Figures 10 and 12. The rate $-\mathcal{R}(t, r_{max})$ decreases over time, and the distance $r_{max}$ at which the rate $-\mathcal{R}(t, r_{max})$ is maximum shifts slightly toward the center of the heating zone. However, $-\mathcal{R}(t, r_{max})$ still exceeds 2–3 GK/s at $t_1 = 100$ ns; see Figures 10a and 12b. The change in cooling rate $-\mathcal{R}(t, r_{max})$ over time for sodium-lime-silicate glass at $r_0 = 1$ µm, $\tau_0 = 10$ µs and the same conditions as in Figure 3 is presented in Table 3. The rate $\mathcal{R}(t_1, r)$ depending on the distance $r$ at the periphery of the hot zone for different $t_1$ is shown in Figure 13.

**Table 3.** Cooling rate $-\mathcal{R}(t, r)$ for sodium-lime-silicate glass at $r_0 = 1$ μm.

| Time $t_1$ ns | Distance $r_{max}$ μm | Cooling Rate $-\mathcal{R}(t_1, r_{max})$ GK/s |
|---|---|---|
| 0.5 | 0.985 | 410 |
| 1 | 0.98 | 170 |
| 5 | 0.94 | 31 |
| 10 | 0.92 | 16 |
| 50 | 0.80 | 4 |
| 100 | 0.70 | 2.3 |

Thus, since the local structure of the material strongly depends on the local cooling rate $-\mathcal{R}(t, r)$, the local structure of the material at the periphery of the hot zone changes significantly relative to the original glass matrix. This modification of glass practically stops as soon as the local region is cooled below the glass transition temperature $T_g$. Thus, the modification process occurs until the temperature $T(t, r_1)$ in the local region at $r = r_1$ drops below $T_g$. For example, consider the cooling curves calculated at the periphery of the hot zone for borosilicate ($T_g$ = 900 K [62]) and sodium-lime-silicate glasses ($T_g$ = 840 K [56]); see Figures 10b and 14, respectively. It should be noted that the effect associated with the time dispersion of the dynamic heat capacity is significant well above the glass transition temperature $T_g$. Indeed, in the case of $r_0 = 1$ μm, the modification process occurs within approximately 80 ns at $r_1 = 0.95$ μm and $\tau_0 \geq 1$ μs; see Figure 14a. However, without taking into account the time dispersion of the dynamic heat capacity (at $\tau_0 = 0$), this process occurs in less than 20 ns; see Figure 14a. In the case of $r_0 = 2$ μm, the modification process occurs during a longer period of about 300 ns at $r_1 = 1.98$ μm and $\tau_0 > 1$ μs; see Figure 10b. However, without taking into account the time dispersion of the dynamic heat capacity (at $\tau_0 = 0$), this process occurs in less than 50 ns; see Figure 10b.

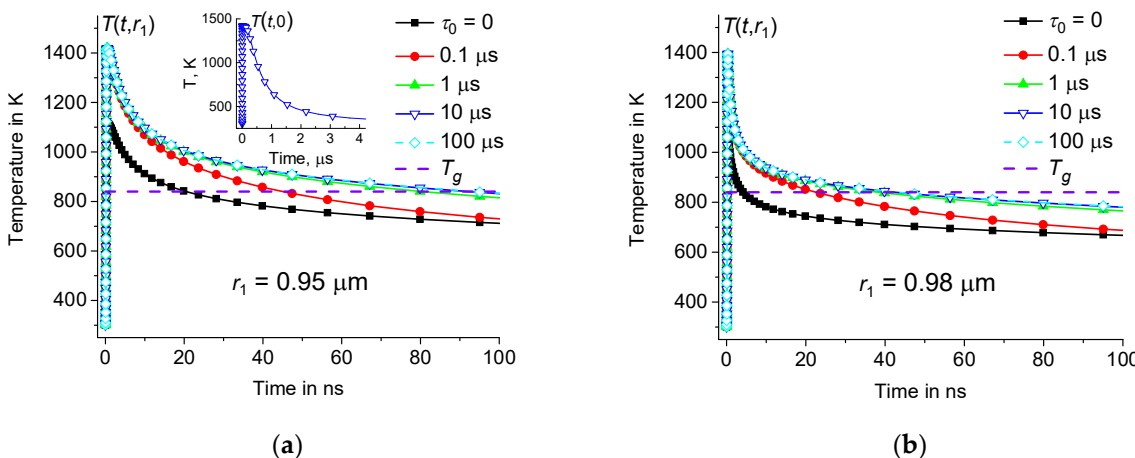

**Figure 14.** Time dependence of $T(t, r_1)$ at the periphery of the hot zone at $r_1 = 0.95$ μm (**a**) and $r_1 = 0.98$ μm (**b**) at the same parameters as in Figure 3 ($\tau_0 = 0$, 0.1 μs, 1 μs, 10 μs, and 100 μs—squares, circles, triangles up, triangles down, and diamonds). Inset (**a**) shows the time dependence of $T(t, 0)$ in the center of heating zone (at $\tau_0 = 10$ μs).

To summarize, we can conclude that the maximum cooling rate exists in the outer regions of the heating zone in the shell of about $(0.97 \pm 0.02) r_0$. The cooling rate reaches several hundred GK/s in the first nanosecond after the laser pulse. Then, the cooling rate decreases. However, even 100 ns after the laser pulse, $-\mathcal{R}(t, r)$ in the outer shell of the hot zone still exceeds 1 GK/s. It is noteworthy that in the center of the heating zone, the cooling rate is three orders of magnitude less than $-\mathcal{R}(t, r)$ at the periphery. Since regions quenched at different cooling rates have different physical properties, strong gradients

of physical properties should exist in the material predominantly at a distance of about $0.9r_0$ from the center of the heating zone. This effect is significantly enhanced by the time dispersion of the dynamic heat capacity.

In conclusion, let us briefly discuss laser microprocessing in the cumulative heating mode. Typically, cumulative heating is achieved at a laser pulse repetition rate of hundreds of kHz or more. Usually, this mode operates at frequencies from 200 kHz to 10 MHz [3]. Indeed, the temperature $T(t, 0)$ in the center of the hot zone relaxes in a time of the order of $r_0^2/D_0$, which for the glasses under consideration is about 4 µs at $r_0 = 1$ µm; see inset in Figure 14a. Thus, in this case, the cumulative effect can be achieved at a frequency of about 250 kHz or more. Note that the cumulative effect leads to stable average heating of the material in the heating zone. Thus, the effect of laser pulse repetition can be approximately simulated by shifting the local temperature distribution $T(t, r)$ upward by a certain temperature difference $\Delta T$, which increases with the increasing pulse repetition frequency. It is clear that with increasing $\Delta T$, the modification process will occur over a longer time interval—see Figure 14—where $T(t, r_1)$ should be shifted upward by the amount $\Delta T$. Then, $T(t, r_1)$ in the local region at $r = r_1$ will fall below $T_g$ at a cooling rate lower than at zero repetition frequency (at $\Delta T = 0$). Consequently, local gradients in the physical properties of locally modified glass will be smoothed out due to cumulative heating. The effect of the repetition rate will be discussed in more detail in a separate article.

## 5. Discussion

The local temperature distribution $T(t, r)$ and the local cooling rate $-\mathcal{R}(t, r)$ determine the local structure of the glass matrix during laser-induced microstructuring. Thus, knowledge of the dynamics of the local temperature distribution $T(t, r)$ is very important for applications associated with local laser heating, for example, for the femtosecond laser microstructuring of glasses. However, as shown in this article, the dynamics of the local temperature distribution $T(t, r)$ significantly depends on the time dispersion of the dynamic heat capacity $c_{dyn}(t)$ of the glass matrix. This work proposes a method for the analytical calculation of $T(t, r)$ and $\mathcal{R}(t, r)$ for glass-forming materials under local fast laser excitations. It is shown that the dynamics of the local temperature distribution $T(t, r)$ caused by a laser pulse can be described by the heat equation with dynamic heat capacity $c_{dyn}(t)$. This equation has an analytical solution for a spherically symmetric boundary value problem. Using this analytical solution, we obtained the temperature distribution $T(t, r)$ and local cooling rate $-\mathcal{R}(t, r)$ for the thermal parameters of borosilicate and sodium-lime-silicate glasses. In fact, this analytical solution depends on the following relaxation parameters of the dynamic heat capacity $c_{dyn}(t)$ of the material: relaxation time $\tau_0$ and the distribution function $H(\tau_0)$, which can be obtained from broadband heat capacity spectroscopy. However, as shown in this article, the shape of the distribution function $H(\tau_0)$ has little effect on $T(t, r)$ and $\mathcal{R}(t, r)$, since with increasing $\tau_0$, the influence of the time dispersion of the dynamic heat capacity $c_{dyn}(t)$ reaches saturation. It should be noted that the influence of the time dispersion of the dynamic heat capacity is most pronounced at the beginning of the process, when $t$ changes on a nanosecond time scale. We found that this effect is significant already at a $\tau_0$ of about 0.1 µs, increases with increasing $\tau_0$, and reaches saturation at $\tau_0$ above $r_0^2/D_0$. Thus, the influence of the time dispersion of dynamic heat capacity on the temperature distribution $T(t, r)$ can be calculated for a fixed, sufficiently large $\tau_0$ (in fact, for $\tau_0$ on the order of ten or several tens of microseconds, depending on the radius of the heating zone $r_0$). As expected, the results obtained are very similar for different $r_0$ values. However, as the size of the heating zone increases, the laser pulse energy $E_p$ must increase in proportion to the volume $V_0$ of the heating zone in order to obtain the same amplitude of the local thermal response.

It has been established that the temperature distribution $T(t, r)$ and local cooling rate $-\mathcal{R}(t, r)$ are not affected by the heating pulse duration $\tau_p$ or the shape of the heating pulse. It should be noted that the temperature perturbation $\delta T(t) = T(t, 0) - T_{in}$ in the center of

the heating zone does not relax as $\delta T \sim t^{-3/2}$, in contrast to the often-used estimate based on the fundamental solution of the Fourier heat equation. In fact, the heating pulse cannot be considered as an instantaneous point source of heat. Thus, it is necessary to integrate the thermal response over the time interval $(0, \tau_p)$ and the volume of the heating zone, and then the resulting correct solution $T_{FS}(t, r)$ completely coincides with the temperature distribution $T(t, r)$ found in this article for $\tau_0 = 0$. In addition, the solution found in this article can be used for materials with dynamic heat capacity $c_{dyn}(t)$ at $\tau_0 \neq 0$.

We found that the rate of temperature change $\mathcal{R}(t, r)$ is greatest near the periphery of the hot zone. The cooling rate reaches a maximum of $-\mathcal{R}(t, r_{max})$ when $r_{max}$ is slightly less than $r_0$. It turns out that ultrafast cooling of the material occurs at a rate of more than $10^{11}$ K/s in a thin shell in the region from $0.95 r_0$ to $0.99 r_0$. However, in the center of the heating zone, the cooling rate is three orders of magnitude lower than at the periphery. Thus, the local structure of glass after laser modification in the center of the hot zone should differ significantly from the structure at the periphery in a thin shell $\Delta r$ of about 0.05 μm, which is consistent with experiments [19,33,35,39,45,48–52,85]. In fact, the experimentally observed thin shell at the periphery of the laser beam focus is visually about $\Delta r / r_0 \approx 10\%$; i.e., $\Delta r$ is about 0.5 μm at $r_0 \approx 5$ μm [19,33,35]. Of course, the visual size $\Delta r$ is smoothed out due to the diffraction limit of optical images of laser-induced microbeads. Indeed, for such laser-induced microbeads, the shell size $\Delta r$ is about 0.05 μm in high-resolution scanning electron microscopy images [49,85]. In addition, the shell size $\Delta r$ is smoothed due to the high-frequency repetition of laser pulses during laser processing in the cumulative heating mode.

The rate $-\mathcal{R} \approx 10^{11}$ K/s of ultrafast cooling of the material in a thin shell at the periphery of the heating zone is sufficient to freeze the liquid structure corresponding to a temperature $T_f$ that is significantly higher than the initial temperature $T_{in}$. In fact, the difference $\Delta T = T_f - T_{in}$ can be on the order of several hundred K. Indeed, the material at the periphery of the heating zone is locally quenched-in within a time interval $\Delta t$ of about 10–100 ns; see Figure 14. In this case, the liquid structure of the material should remain frozen, since the distance $\sqrt{\Delta t D_{SD}}$ of self-diffusion of glass molecules does not exceed the order of several nanometers with the self-diffusion coefficient $D_{SD}$ of the order of $10^{-9}$ m$^2$/s [64]. Suppose the material is locally heated and then rapidly cooled to the initial temperature of $T_{in}$ of the matrix material. Thus, if a rapidly cooled material has a locally frozen density $\rho\left(T_f\right)$ corresponding to a certain temperature $T_f = T_{in} + \Delta T$, then the local density change $\Delta \rho / \rho$ is about $\alpha_V \Delta T$, where $\alpha_V$ is the volumetric thermal expansion coefficient. For example, for borosilicate glasses, $\Delta \rho / \rho$ is about 0.2% at $\Delta T = 200$ K, $\alpha_V = 3\alpha$, and linear thermal expansion $\alpha = 3.3 \cdot 10^{-6}$ 1/K [60]. Then, the local stress can reach the order of $K_B \alpha_V \Delta T$ (about 65 MPa) with a bulk modulus $K_B = 33$ GPa [60,61]. For sodium-lime-silicate glass, $\Delta \rho / \rho$ is about 0.5% at $\Delta T = 200$ K, $\alpha_V = 3\alpha$, and linear thermal expansion $\alpha = 7.7 \cdot 10^{-6}$ 1/K [59]. Then, the local stress can reach about $K_B \alpha_V \Delta T \approx 250$ MPa with a bulk modulus $K_B = 55$ GPa [55,58]. This local stress can cause a change in the local birefringence on the order of several percent [86]. A similar change in the local refractive index, proportional to $\Delta \rho / \rho$, can be caused by a change in the local density [13]. This estimate is consistent with femtosecond laser anisotropic nanostructuring in transparent materials for optical data storage [3]. Of course, the stress caused by the laser pulse then relaxes over time to some residual value. This residual stress may be the subject of a separate study. In this work, we focus on the unique capability of femtosecond laser microstructuring deep inside transparent glasses. This possibility can be used for optical data storage. In fact, changes in the refractive index with changes in local density $\Delta \rho / \rho$ and changes in the birefringence due to local stress are sufficient for data storage [13]. The locally frozen density $\rho\left(T_f\right)$ depends on the local cooling rate $-\mathcal{R}(t, r)$ near the laser beam focus. The gradients of the local cooling rate $-\mathcal{R}(t, r)$ calculated in this work lead to the gradients of the locally frozen density and optical properties of the material. Such gradients have been observed at the periphery of the laser beam focus [19,33,35,39,45,48–52,85]. Thus, we have good agreement with the experiment. However, such experiments were usually

carried out at different laser pulse repetition rates. In this case, these gradients of physical properties should be smoothed relative to the case of the single laser pulse considered in this work. A future direction of research may be the influence of laser pulse repetition rate on local gradients of locally frozen density and other physical properties.

It is noteworthy that the effect associated with the time dispersion of the dynamic heat capacity is very significant both in the center and at the periphery of the heating zone. In fact, the maximum cooling rate is approximately 2.5 times greater at $\tau_0 \neq 0$ than at $\tau_0 = 0$ both at the periphery and in the center of the heating zone; see Figure 11.

It is worth noting that at the beginning of the cooling process at the periphery of the hot zone, the shape of the distribution function $H(\tau_0)$ does not affect the cooling rate $-\mathcal{R}(t,r)$. In fact, even at a very short relaxation time $\tau_0$, the influence of the time dispersion of the dynamic heat capacity is very significant at the beginning of the cooling process. The cooling rate decreases over time $t$; however, in the outer regions of the heating zone, the cooling rate $-\mathcal{R}(t,r)$ still exceeds 2–3 GK/s at $t$ about 100 ns. Regions quenched at different cooling rates have different physical properties. Thus, strong gradients of physical properties should exist in the material, mainly at a distance of about $0.9r_0$ from the center of the heating zone. It is worth noting that the time dispersion of the dynamic heat capacity significantly enhances this effect. In addition, the effect associated with the time dispersion of the dynamic heat capacity is significant even well above the glass transition temperature $T_g$. It turns out that the duration of the modification process strongly depends on the relaxation time $\tau_0$ of the dynamic heat capacity $c_{dyn}(t)$. Thus, the rate of local cooling and gradients of physical properties of locally modified glass strongly depend on the relaxation time of the dynamic heat capacity $c_{dyn}(t)$. Finally, it is worth noting that the gradients of physical properties in areas quenched at different cooling rates are smoothed out by cumulative heating when processed in the cumulative heating mode.

In this article, we solved a problem for spherical geometry, which is closely related to data storage technologies. There are a number of future directions that are worth developing. A future research direction could be the problem with cylindrical geometry, which is closely related to laser waveguide technologies. The cumulative heating mode at different laser pulse repetition rates and laser beam scanning speeds also deserves to be studied.

## 6. Conclusions

To summarize, it can be emphasized that during the laser microstructuring of glass-forming materials, the local temperature distribution $T(t,r)$ and the local cooling rate significantly depend on the time dispersion of the dynamic heat capacity $c_{dyn}(t)$ of the glass matrix. The rate of temperature change $\mathcal{R}(t,r)$ is at a maximum near the periphery of the heating zone. However, the effect associated with the time dispersion of the dynamic heat capacity is very significant both in the center and at the periphery of the heating zone. The effect associated with the time dispersion of the dynamic heat capacity is significant even well above the glass transition temperature $T_g$. It turns out that in the thin shell of the heating zone, the ultrafast cooling of the material occurs at a rate of more than $10^{11}$ K/s, and strong gradients of physical properties must exist in the material, mainly in a thin shell around the heating zone. The time dispersion of the dynamic heat capacity significantly enhances this effect. However, these gradients in physical properties should be smoothed out when processed in the cumulative heating mode. Further directions of research may be related to the consideration of the cumulative heating regime for various geometries. The results of this work can be useful for a better understanding and optimization of technologies associated with laser-induced microstructuring of glasses.

**Author Contributions:** Conceptualization, A.M.; formal analysis, A.M.; methodology, A.M.; supervision, C.S.; visualization, A.M.; writing—original draft, A.M.; writing—review and editing, C.S. All authors have read and agreed to the published version of the manuscript.

**Funding:** This research received no external funding.

**Institutional Review Board Statement:** Not applicable.

**Informed Consent Statement:** Not applicable.

**Data Availability Statement:** The raw data supporting the conclusions of this article will be made available by the authors on request.

**Acknowledgments:** A.M. acknowledges the administrative and technical support of the Prokhorov General Physics Institute of the Russian Academy of Sciences.

**Conflicts of Interest:** The authors declare no conflicts of interest.

## Nomenclature

Latin Symbols

| | |
|---|---|
| $B$ and $T_0$ | Parameters of the VFTH equation (K) |
| $c_S$ | Heat capacity of the solid material (J·kg$^{-1}$·K$^{-1}$) |
| $c_L$ | Heat capacity of the liquid material (J·kg$^{-1}$·K$^{-1}$) |
| $c_{in}$, $c_0$ | Initial and equilibrium heat capacities (J·kg$^{-1}$·K$^{-1}$) |
| $c_{dyn}(t)$ | Dynamic heat capacity (J·kg$^{-1}$·K$^{-1}$) |
| $D_0$ | Thermal diffusivity $\lambda/\rho c_0$ (m$^2$·s$^{-1}$) |
| $E_p$ | Energy of laser pulse (nJ) |
| $f_0$ | Parameter of the VFTH equation (Hz) |
| $F(t)$ | Time dependence of the pulse power (dimensionless) |
| $G\left(t, \vec{r}\right)$ | Fundamental solution of the Fourier heat equation (m$^{-3}$) |
| $H(\tau_0)$ | Distribution function (s$^{-1}$) |
| $K_B$ | Bulk modulus (GPa) |
| $\mathcal{R}(t,r)$ | Rate of temperature change $\mathcal{R}(t,r) = \partial T(t,r)/\partial t$ (K·s$^{-1}$) |
| $-\mathcal{R}(t,r)$ | Cooling rate (K·s$^{-1}$) |
| $R_0$ | Parameter of the boundary value problem (μm) |
| $r_0$ | Heating zone radius (μm) |
| $r$ | Distance from the center of the hot zone (μm) |
| $t$ | Time (s) |
| $T_{FS}(t,r)$ | Temperature distribution obtained from fundamental solution (K) |
| $T(t,r)$ | Local temperature distribution (K) |
| $T_{AV}(t,r)$ | Temperature distribution averaged using $H(\tau_0)$ (K) |
| $T_g$ | Glass transition temperature (K) |
| $T_{in}$ | Initial temperature (K) |
| $T_{int}$ | Intermediate temperature (K) |
| $T_{max}$ | Maximum value of $T(t,r)$ (K) |
| $V_0$ | Heating zone volume (m$^3$) |
| $V$ | Specific volume (m$^3$·kg$^{-1}$) |
| $V_L$ | Longitudinal speed of sound (m·s$^{-1}$) |

Greek Symbols

| | |
|---|---|
| $\alpha$ | Linear thermal expansion (K$^{-1}$) |
| $\alpha_V$ | Volumetric thermal expansion coefficient (K$^{-1}$) |
| $\beta$ | Parameter of the Kohlrausch relaxation law (dimensionless) |
| $\gamma_n$ | Relaxation parameter (dimensionless) |
| $\varepsilon_0$ | Coefficient $\varepsilon_0 = (c_0 - c_{in})/c_0$ (dimensionless) |
| $\lambda$ | Thermal conductivity (W·m$^{-1}$·K$^{-1}$) |
| $\mu_n$ | Relaxation parameter (dimensionless) |

| | |
|---|---|
| $\rho$ | Density (kg·m$^{-3}$) |
| $\tau_0$ | Debye relaxation time (μs) |
| $\tau_K$ | Kohlrausch relaxation time (μs) |
| $\tau_{laser}$ | Duration of laser pulse (fs) |
| $\tau_n$ | $n$th relaxation time (μs) |
| $\tau_p$ | Duration of heating pulse (ps) |
| $\Phi(t, r)$ | Volumetric heat flux density (W·m$^{-3}$) |
| $\Phi_0$ | Volumetric heat flux density (W·m$^{-3}$) |
| $\Phi_n$ | $n$th Fourier component (W·m$^{-2}$) |
| $\psi_n(t)$ | $n$th Fourier component (K·m) |
| $\omega$ | Temperature modulation frequency (rad·s$^{-1}$) |

**Appendix A**

Equation (7) can be converted into Equation (A1) by replacing $r(T(t, r) - T_{in})$ with $U(t, r)$:

$$U\prime(t, r) - D_0 \partial^2 U / \partial r^2 = \frac{r\Phi(r)F(t)}{\rho c_0} + \varepsilon_0 \frac{\partial}{\partial t} \int_0^t exp\left(-\frac{t-\tau}{\tau_0}\right) U\prime(\tau, r) d\tau, \qquad \text{(A1)}$$

Thus, we obtain a one-dimensional problem with uniform boundary and initial conditions: $U(t, 0) = 0$, $U(t, R_0) = 0$, and $U(t, r) = 0$ at $t \leq 0$.

This boundary value problem is satisfied by the following series:

$$U(t, r) = \sum\nolimits_{n=1} \psi_n(t) sin(\pi n r / R_0), \qquad \text{(A2)}$$

where the functions $\psi_n(t)$ are the solutions of Equation (9) [69]. Thus, for a heat source uniformly distributed in a volume of radius $r_0$ with power density $\Phi_0$, and for the heating pulse $\Phi_0 sin(\pi t / \tau_p)$, acting during the time interval $\tau_p$, we obtain solutions $\psi_n(t)$ to Equation (9) represented by Equations (A3) and (A4) for $0 \leq t \leq \tau_p$ and $\tau_p < t$, respectively.

$$\psi_n(t) = \frac{\Phi_n}{\rho c_0} \frac{\tau_n \gamma_n \mu_n}{(\gamma_n - \mu_n)} \left[ \frac{(\gamma_n \tau_0 - 1)\left[\gamma_n sin(\pi t / \tau_p) + \frac{\pi}{\tau_p}(exp(-\gamma_n t) - cos(\pi t / \tau_p))\right]}{(\gamma_n)^2 + \left(\frac{\pi}{\tau_p}\right)^2} + \frac{(1 - \mu_n \tau_0)\left[\mu_n sin(\pi t / \tau_p) + \frac{\pi}{\tau_p}(exp(-\mu_n t) - cos(\pi t / \tau_p))\right]}{(\mu_n)^2 + \left(\frac{\pi}{\tau_p}\right)^2} \right], \qquad \text{(A3)}$$

$$\psi_n(t) = \frac{\Phi_n}{\rho c_0} \frac{\pi \tau_n \gamma_n \mu_n}{\tau_p(\gamma_n - \mu_n)} \left[ \frac{(\gamma_n \tau_0 - 1)\left[exp(-\gamma_n t) + exp(\gamma_n(\tau_p - t))\right]}{(\gamma_n)^2 + \left(\frac{\pi}{\tau_p}\right)^2} + \frac{(1 - \mu_n \tau_0)\left[exp(-\mu_n t) + exp(\mu_n(\tau_p - t))\right]}{(\mu_n)^2 + \left(\frac{\pi}{\tau_p}\right)^2} \right], \qquad \text{(A4)}$$

where $-\gamma_n$ and $-\mu_n$ are the roots of the polynomial $(1 - \varepsilon_0)p^2 + p\left(\tau_n^{-1} + \tau_0^{-1}\right) + \tau_n^{-1}\tau_0^{-1}$. The parameters $\gamma_n$ and $\mu_n$ are real, positive, and $(\gamma_n - \mu_n) \neq 0$ for $0 < \varepsilon_0 < 1$ and $0 < \tau_n$, $\tau_0$. The series in Equation (8) converges as $1/n^2$ for $\tau_p < t$. In fact, it is enough to calculate the sum in Equation (8) to about a thousand terms to obtain a result with an error of less than 1%.

Similarly, for a heating pulse of a square shape with power density $\Phi_0$, acting on a time interval $\tau_p$, we obtain $\psi_n(t) = \phi_n(t)$ and $\psi_n(t) = \left[\phi_n(t) - \phi_n(\tau_p - t)\right]$ for $\tau_p < t$ and $0 \leq t \leq \tau_p$, respectively, where $\phi_n(t)$ are represented by Equation (A5):

$$\phi_n(t) = \frac{\Phi_n}{\rho c_0} \tau_n \left[1 + \frac{\tau_0 \gamma_n \mu_n (exp(-\mu_n t) - exp(-\gamma_n t))}{(\gamma_n - \mu_n)} + \frac{\mu_n exp(-\gamma_n t) - \gamma_n exp(-\mu_n t)}{(\gamma_n - \mu_n)}\right]. \qquad \text{(A5)}$$

Note that the solutions $\phi_n(t)$ continuously transform into solutions of the classical Fourier heat equation: $\phi_n(t) \to \frac{\Phi_n}{\rho c_0} \tau_n [1 - exp(-t/\tau_n)]$ as $\varepsilon_0 \to 0$ or/and $\tau_0 \to 0$.

Similarly, the functions $\psi_n(t)$ represented by Equations (A3) and (A4) continuously transform into solutions of the classical Fourier heat equation as $\varepsilon_0 \to 0$ or/and $\tau_0 \to 0$—see Equations (A6) and (A7)—for $0 \leq t \leq \tau_p$ and $\tau_p < t$, respectively.

$$\psi_n(t) = \frac{\Phi_n}{\rho c_0} \frac{\frac{1}{\tau_n} sin(\pi t / \tau_p) + \frac{\pi}{\tau_p}\left[exp(-t/\tau_n) - cos(\pi t / \tau_p)\right]}{\left(\frac{1}{\tau_n}\right)^2 + \left(\frac{\pi}{\tau_p}\right)^2}, \qquad \text{(A6)}$$

$$\psi_n(t) = \frac{\Phi_n}{\rho c_0} \frac{\pi}{\tau_p} \frac{\left[ exp(-t/\tau_n) + exp((\tau_p - t)/\tau_n)) \right]}{\left(\frac{1}{\tau_n}\right)^2 + \left(\frac{\pi}{\tau_p}\right)^2}. \tag{A7}$$

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
