# Peer review of "Temperature Relaxation in Glass-Forming Materials under Local Fast Laser Excitations during Laser-Induced Microstructuring"

_applsci, doi:10.3390/app14031076_

Round 1

Reviewer 1 Report

Comments and Suggestions for Authors

The work is an interesting presentation of the thermal considerations in glasses invoking both local structure, composition and dynamic consideration.  The model would seem to be rather effective in modelling the temperature distribution of glasses during the cooling process.  This of course has implication for volumetric changes and the consequent stress distributions that arise. The role of stress distribution and shrinkage of glass during processing is rather important in a range of industrial applications. 

Glazes, specialty glasses, coatings and optics. Optics has been mentioned in the paper. However further examination of the role of temperature and stress distribution on processing and performance of specialty glass and glaze products would be beneficial here. 

To summarize, the manuscript can be accepted, however further comments are requested regarding the applications and significance of this work. 

Reviewer 2 Report

Comments and Suggestions for Authors

It is an interesting contribution on the theoretical analysis of the heat induced changes in glasses. I suggest publishing the paper subject to several issues being clarified by the authors:

1. Are the analytical solutions presented a part of novel contribution made by the paper or they have been presented somewhere earlier in other publications?

2. With respect to formula 12 - is obtaining this formula actually consisting in applying Greens function method? If so it seems to be missing in the text? Also with formula 12 - it involves integration - how this integration was performed? Was it a numerical approach? If so, how it was done? If analytical then what is the final formula?

3. Considering the context of applied sciences I would expect some direct reference to experimental results. This could be done by calibrating the model with some experimental results or by comparing the modelling results with experimental results. Otherwise the simulations may predict trends that are well of the actual true behavior. Could the authors at least comment on this problem in the text by saying how far they believe they results might differ from reality?
